# Triglyceride-Catabolizing *Lactiplantibacillus plantarum* GBCC_F0227 Shows an Anti-Obesity Effect in a High-Fat-Diet-Induced C57BL/6 Mouse Obesity Model

**DOI:** 10.3390/microorganisms12061086

**Published:** 2024-05-27

**Authors:** Jinwook Kim, Seong-Gak Jeon, Min-Jung Kwak, So-Jung Park, Heeji Hong, Seon-Bin Choi, Ji-Hyun Lee, So-Woo Kim, A-Ram Kim, Young-Kyu Park, Byung Kwon Kim, Bo-Gie Yang

**Affiliations:** Research Institute, GI Biome Inc., Seongnam-si 13201, Republic of Korea; jinwook152@gi-biome.com (J.K.); sgjeon@gi-biome.com (S.-G.J.); minjung.kwak@gi-biome.com (M.-J.K.); sjpark@gi-biome.com (S.-J.P.); heeji.hong@gi-biome.com (H.H.); sbchoi@gi-biome.com (S.-B.C.); jhlee@gi-biome.com (J.-H.L.); sw.kim@gi-biome.com (S.-W.K.); arkim@gi-biome.com (A.-R.K.); ykpark@gi-biome.com (Y.-K.P.); bkkim@gi-biome.com (B.K.K.)

**Keywords:** *Lactiplantibacillus plantarum*, triglyceride, α/β hydrolases, lipase, adiponectin, adipose tissue, obesity

## Abstract

Given the recognized involvement of the gut microbiome in the development of obesity, considerable efforts are being made to discover probiotics capable of preventing and managing obesity. In this study, we report the discovery of *Lactiplantibacillus plantarum* GBCC_F0227, isolated from fermented food, which exhibited superior triglyceride catabolism efficacy compared to *L. plantarum* WCSF1. Molecular analysis showed elevated expression levels of α/β hydrolases with lipase activity (abH04, abH08_1, abH08_2, abH11_1, and abH11_2) in *L. plantarum* GBCC_F0227 compared to *L. plantarum* WCFS1, demonstrating its enhanced lipolytic activity. In a high-fat-diet (HFD)-induced mouse obesity model, the administration of *L. plantarum* GBCC_F0227 mitigated weight gain, reduced blood triglycerides, and diminished fat mass. Furthermore, *L. plantarum* GBCC_F0227 upregulated adiponectin gene expression in adipose tissue, indicative of favorable metabolic modulation, and showed robust growth and low cytotoxicity, underscoring its industrial viability. Therefore, our findings encourage the further investigation of *L. plantarum* GBCC_F0227’s therapeutic applications for the prevention and treatment of obesity and associated metabolic diseases.

## 1. Introduction

Driven by the adoption of Western dietary patterns, characterized by the increased consumption of high-fat diets (HFDs) with low dietary fiber, the incidence of obesity is rapidly increasing worldwide. Obesity causes chronic metabolic diseases such as type 2 diabetes, dyslipidemia, and nonalcoholic fatty liver disease. Excess energy intake from HFDs is mainly stored in adipose tissues, and as obesity progresses, adipocytes expand in size and fat mass increases. Adipose tissue is also involved in metabolic, hormonal, and immune processes, whose products and reactions can affect other organs, playing an important role in whole-body homeostasis [1]. Fat accumulation elicits inflammatory responses in adipose tissue, as demonstrated by elevated levels of inflammatory immune cells such as IFN-γ^+^ CD4^+^ T cells, CD8^+^ T cells, and M1 macrophages alongside reduced anti-inflammatory immune cell populations, including Foxp3^+^CD4^+^ regulatory T cells and M2 macrophages [2,3,4,5,6]. This results in insulin resistance and glucose intolerance [7]. In addition, adipose tissue secretes various adipokines such as chemokines, cytokines, and hormones, which are involved in the regulation of energy homeostasis and inflammation [8]. Notably, adiponectin improves obesity-related metabolic symptoms, enhancing insulin sensitivity and reducing inflammatory cytokines such as TNF-α [9]. Conversely, resistin has the opposite effect, exacerbating these symptoms. Obesity diminishes the expression of anti-inflammatory adipokines like adiponectin while augmenting pro-inflammatory adipokine expression such as resistin. In particular, adiponectin can both exist in the form of full-length adiponectin oligomers, which possess both a collagen-like fibrous domain and a C1q-like globular domain, and as globular adiponectin, which contains a C1q-like globular domain alone [10]. Full-length adiponectin primarily binds to the adiponectin receptor AdipoR2, whereas globular adiponectin binds to AdipoR1 [11]. The tissue expression patterns of AdipoR1 and AdipoR2 are different; the expression of AdipoR1 is ubiquitous, but is particularly high in skeletal muscles, while the expression of AdipoR2 is predominant in the liver [11]. Adiponectin signaling increases fatty acid oxidation in both the liver and skeletal muscle. In skeletal muscle, it promotes glucose uptake, contributing to glucose consumption, while in the liver, it inhibits glucose production by reducing gluconeogenesis [12]. These actions collectively improve lipid and glucose metabolism, thereby increasing insulin sensitivity and lowering blood glucose levels. Considering the above, adiponectin has emerged as a promising treatment target for obesity-related metabolic diseases [8].

Many studies have demonstrated a close association between the gut microbiome and obesity [13,14,15]. For example, genetically identical twins have occasionally exhibited discordant obesity attributed to variations in their gut microbiome. Fecal transplantations into germ-free (GF) mice revealed that mice receiving feces from obese twins became obese, while those receiving feces from lean twins did not [16,17], while the transplantation of fecal matter from obese and lean mice into GF mice resulted in significantly greater weight gain in mice that received feces from their obese counterparts [13]. These results underscore the pivotal role of the gut microbiome in the development of obesity and its potential for transmitting obesity phenotypes. Additionally, HFD-induced obesity has been linked to reduced gut microbiome abundance and diversity, accompanied by a decrease in butyrate-producing bacteria and an increase in harmful bacteria, which lead to dysbiosis [18,19]. Such changes in the gut microbiome increase the permeability of the gut to endotoxins such as LPSs (Lipopolysaccharides) and in turn trigger inflammatory responses which are involved in adiposity, insulin resistance, and dyslipidemia [15,20]. Obesity also increases the ratio of *Firmicutes* to *Bacteroidetes*, and it has been reported that *Lactobacillus* spp., a member of *Firmicutes*, shows a positive correlation with the inflammatory marker high-sensitivity C-reactive protein (hs-CRP), especially in obese children [21]. However, in another study, no significant changes in *Lactobacillus* spp. due to obesity were observed [22]. Rather, there are reports that various *Lactobacillus* spp. strains have anti-obesity effects [12]. Thus, an expanding body of research aims to utilize the gut microbiome to develop preventive and therapeutic interventions against obesity and related metabolic diseases [23,24].

Triglycerides are the major dietary lipids, accounting for nearly 90–95% of the energy from fat, and are the main contributor to HFD-induced obesity [25]. Therefore, in this study, we aim to discover new anti-obesity strains that efficiently catabolize triglycerides and inhibit their absorption. The exploration of novel gut microbiome strains demands considerable time and resources to ensure their safety and facilitate mass production for industrial development. On the other hand, strains sourced from traditionally consumed lactic acid bacteria species entail relatively lower time and effort investments. In this context, our investigation identified a traditional lactic acid bacteria strain proficient in catabolizing triglycerides and blocking the absorption of fats. Specifically, *Lactiplantibacillus plantarum* GBCC_F0227, isolated from pickled cabbages, demonstrated efficacy in suppressing weight gain and reducing blood triglyceride levels in an HFD-induced mouse obesity model. In addition, *L. plantarum* GBCC_F0227 showed robust growth potential and low cytotoxicity, suggesting that it will face minimal obstacles in industrial development.

## 2. Materials and Methods

### 2.1. Isolation and Identification of New Lactic Acid Bacteria

New lactic acid bacteria were isolated from diverse sources, including various fermented foods and human feces. To isolate lactic acid bacteria, homogenized samples were diluted and anaerobically cultured in an MRS agar plate (BD Difco, Franklin Lakes, NJ, USA) at 37 °C for approximately 65 h. For the anaerobic culture, anaerobic conditions were maintained using an anaerobic chamber (Whitley A45 workstation; Don Whitley Scientific, Bingley, UK) supplied with a gas mixture (N_2_:H_2_:CO_2_ = 90:5:5). Individual colonies were subcultured on a new MRS agar plate for one day and identified by the amplification of the 16S rRNA gene through a PCR reaction using bacterial universal primers and sequencing. To ensure purity, each selected colony was streaked twice on new MRS agar plates and cultured. The resulting colonies were collected in 1 mL of 30% glycerol, mixed, and stored at −70 °C. These newly isolated and identified strains were registered in the GI Biome Culture Collection (GBCC).

### 2.2. Measurement of Triglyceride in Culture Medium

Newly isolated lactic acid bacteria were individually cultured in MRS broth medium at 37 °C for 16 h under anaerobic conditions. Following incubation, the culture medium was centrifuged at 10,000× *g* for 10 min, and the supernatant was collected and passed through a 0.22 μm syringe filter to obtain the cell-free supernatant (CFS). Then, 10 μL of the CFS was dropped onto a TG-P III colorimetric method slide, and the triglyceride concentration was measured using a DRI-CHEM NX700 clinical chemistry analyzer (FUJIFILM, Tokyo, Japan). *L. plantarum* WCFS1 (ATCC BAA-793), *Lacticaseibacillus rhamnosus* Gorbach-Goldin (LGG; KCTC 5033), *Lacticaseibacillus plantarum* KCTC 3108, *Limosilactobacillus fermentum* KCTC 3112, and *Lactobacillus gasseri* KCTC 3163 were used as controls. *L. plantarum* WCFS1 was purchased from the ATCC (American Type Culture Collection, Manassas, VA, USA), and the remaining strains were purchased from the KCTC (Korean Collection for Type Cultures, Jeongeup-si, Republic of Korea).

### 2.3. Culture Conditions of L. plantarum GBCC_F0227

For the growth curve analysis, *L. plantarum* GBCC_F0227 was inoculated into the MRS medium to achieve an optical density value of 0.1 at 600 nm (OD_600_) and cultured at 37 °C under anaerobic or aerobic conditions. Each culture was conducted independently in triplicate, and OD_600_ values were measured every 6 h throughout the culturing period. For anaerobic culture, a Whitley A45 workstation (Don Whitley Scientific, Bingley, UK) was used. For aerobic culture, a JEIOTECH incubator (ISS-3075R; Daejeon, Republic of Korea) was used without agitation. For electron microscopy imaging, *L. plantarum* GBCC_F0227 was anaerobically cultured at 37 °C for 16 h, followed by fixation in 0.05 M sodium cacodylate buffer containing 2% glutaraldehyde and 2% paraformaldehyde. All subsequent procedures were performed in the National Instrumentation Center for Environmental Management (NICEM; Seoul, Republic of Korea).

### 2.4. Acid Resistance Test

To examine acid resistance under various pH conditions, the pH of the MRS medium was adjusted to a specific pH using HCl and NaOH solution. Then, *L. plantarum* GBCC_F0227 or LGG was inoculated into the medium. Each strain was cultured anaerobically at 37 °C, and the number of viable cells was measured at 2 h and 4 h of culture using the spread plate method. In another experiment, artificial gastric fluid (Biochemazone, Leduc, AB, Canada) was prepared at a final pH adjusted to 3.0 with NaOH and artificial intestinal fluid (Biochemazone, Leduc, AB, Canada) was prepared by the addition of 0.1% (*w*/*v*) of pancreatin (Sigma, St. Louis, MO, USA) and 0.3% (*w*/*v*) of bile salts (Oxoid, Cheshire, UK) and the final pH was adjusted to 8.0 with NaOH. *L. plantarum* GBCC_F0227 and LGG cultured overnight in MRS broth were harvested, washed, and resuspended in PBS. They were inoculated in artificial gastric fluid to achieve an optical density value of 1 at OD_600_ and incubated at 37 °C for 2 h. Then, after washing with PBS, cells were resuspended in artificial intestinal fluid and incubated at 37 °C for 4 h. During the entirety of the incubation, the number of viable cells was measured at 2 h intervals. All experiments were conducted under aerobic conditions.

### 2.5. Cytotoxicity Assay

The human colorectal epithelial cell line Caco-2 was purchased from the Korean Cell Line Bank (KCLB, Seoul, Republic of Korea) and cultured in MEM (Welgene, Gyeongsan-si, Republic of Korea) supplemented with 10% fetal bovine serum (Gibco, Waltham, MA, USA), 1% MEM NEAA (MEM Non-Essential Amino Acids Solutions; Welgene, Republic of Korea), and 1% penicillin–streptomycin (Welgene, Republic of Korea) in a CO_2_ incubator at 37 °C. For the cytotoxicity assay, varying concentrations of *L. plantarum* GBCC_ F0227 (1 × 10^7^, 1 × 10^8^, and 1 × 10^9^ cells/well) were added to each well of a 96-well plate containing Caco-2 cells at a concentration of 1 × 10^4^ cells/well. The plate was then incubated in a CO_2_ incubator at 37 °C for 24 h. Subsequently, a lactate dehydrogenase (LDH) assay was performed using CyQUANT LDH Cytotoxicity assay kits (Invitrogen, Waltham, MA, USA) and SpectraMax iD3 ELISA readers (Molecular Devices, San Jose, CA, USA). As per the manufacturer’s protocol, the lysis buffer served as the positive control and sterilized water served as the negative control. The percentage of cytotoxicity was calculated as follows: %Cytotoxicity = [(Compound-treated LDH activity − Spontaneous LDH activity)/(Maximum LDH activity − Spontaneous LDH activity)] × 100.

### 2.6. Whole-Genome Sequencing and Analysis

For whole-genome sequencing, *L. plantarum* GBCC_F0227 was cultured until the exponential growth phase and then harvested using centrifugation (GYLZ-1736R; Labogene, Seoul, Republic of Korea) at 8000 rpm for 5 min. The harvested cells were washed twice with 1× phosphate-buffered saline (PBS) and frozen before the DNA extraction. Genomic DNA was extracted using a MagAttract HMW DNA kit (Qiagen, Hilden, Germany). A library for PacBio sequencing was prepared using the PacBio Express Template Preparation Kit 2.0 (Pacific Bioscience, Menlo Park, CA, USA), while a library for Illumina sequencing was prepared using the Illumina TruSeq Nano DNA Library Prep Kit (Illumina, San Diego, CA, USA). Genome sequencing with each library was performed using PacBio Sequel IIe and Illumina NovaSeq 6000 (DNALink, Seoul, Republic of Korea). The sequencing reads from the Illumina platform were quality-trimmed using Trimmomatic v.0.39 [26]. De novo assembly of the sequencing reads from the PacBio Sequencing platform was carried out using the microbial assembly protocol included in SMRT Link v11.0.0 and polished using the SMRT Link Arrow algorithm [27,28]. The polished genome sequence was validated using Pilon v.1.22 with short reads from the Illumina platform [29]. Protein-coding genes were predicted using Prodigal v.2.6.3 [30], and functional annotation was conducted using BLAST searches against the UniProt [31], Pfam [32], and COG [33] databases. rRNA, tRNA, and other ncRNA elements were predicted using Infernal cmscan v.1.1.1 with the Rfam database [34,35]. For species identification based on genome sequences, the average nucleotide identity values between *L. plantarum* GBCC_F0227 and various type strains were calculated using Jspecies v.1.2.1 [36]. Furthermore, potential virulence factors were predicted by homology searches against the Virulence Factor Database (VFDB) [37].

### 2.7. Identification of Bacterial Enzymes with Lipase Activity Using Comparative Genome Analysis

The genomes of *L. plantarum* WCSF1 and *L. plantarum* DSM 20174 were retrieved from the National Center for Biotechnology Information (NCBI) database. The protein-coding sequences (CDS) of each genome were predicted using Prodigal. Orthologs of CDS from the genomes were identified using the OrthoMCL program [38] with an inflation value of 2.0. Among the core orthologs, enzyme genes predicted to have lipase activity were identified using hmmscan from the HMMER v.3.1b1 package [39] based on the amino acid sequences of the α/β hydrolases obtained from the Lipase Engineering Database [40]. PCR primers targeting specific α/β hydrolases were designed using Primer-BLAST with the CDS extracted from the ortholog dataset [41].

### 2.8. RNA Extraction and Quantitative RT-PCR (qRT-PCR) Analysis

Excised tissues were placed in RNA Later (Invitrogen, MA, USA) and stored at −80 °C until further use. Upon retrieval, the stored tissues were transferred to individual tubes containing 1 mL of the lysis buffer and then homogenized for 10 min using a tissue homogenizer (Retsch, Haan, Germany) equipped with 5 mm stainless steel beads (Qiagen, Germany). For bacterial RNA extraction, bacteria were anaerobically cultured in MRS medium at 37 °C until the OD_600_ value reached 3 or 6. After centrifugation at 10,000× *g* for 10 min, the supernatant was discarded, and the bacterial pellet was resuspended in 1 mL of lysis buffer. Subsequently, the bacterial cells were homogenized for 10 min with a tissue homogenizer (Retsch, Haan, Germany) using a Lysing Matrix E (MP biomedicals, Solana Beach, CA, USA). RNA extraction was performed using the Easy-spin Total RNA Extraction Kit (iNtRON Biotechnology, Seongnam-si, Republic of Korea). cDNA was synthesized from the extracted RNA using a SuPrimeScript cDNA Synthesis Kit (Genetbio, Daejeon, Republic of Korea) according to the manufacturer’s protocol. Oligo dT was used for tissue cDNA synthesis, while random hexamers were used for bacterial cDNA synthesis. The qRT-PCR was performed using SYBR green Master Mix (Bioneer, Daejeon, Republic of Korea) in an Applied Biosystems™ QuantStudio™ 3 Real-Time PCR System (Applied Biosystems, Waltham, MA, USA). The expression levels of specific genes were normalized to the housekeeping genes: GAPDH for tissue and RNA polymerase beta (rpoβ) for bacteria. The primer sequences used for RT-qPCR are detailed in Appendix A.

### 2.9. HFD-Induced Obesity Mouse Model

All animal experiments were conducted at the GI Biome animal facility with Institutional Animal Care and Use Committee (IACUC) approval (approval code: GIB 23-07-009). Five-week-old male C57BL/6 mice were procured from Orient bio (Seongnam-si, Republic of Korea). The mice were housed under specific-pathogen-free (SPF) conditions and maintained in a temperature-controlled environment with a 12 h dark/light cycle. After one week of acclimatization, all mice received daily oral administration of PBS for two weeks to facilitate stress tolerance. Then, the mice were divided into three groups for experimentation: (1) a group on a normal chow diet (NCD; Teklad global 18% protein rodent diet, 2018C; Inotiv, West Lafayette, IN, USA), (2) a group on a high-fat diet (HFD; rodent diet with 60% kcal fat, D12492; Research Diets, New Brunswick, NJ, USA), and (3) a group fed *L. plantarum* GBCC_F0227 in addition to the HFD. Freeze-dried *L. plantarum* GBCC_F0227 was prepared at a concentration of 5.0 × 10^9^ CFU in 200 μL of PBS per mouse and orally administered daily. Mice in the control groups received 200 μL of PBS. The body weight of the mice was measured weekly. Body fat and lean mass were assessed using a minispec LF50 body composition analyzer (Bruker, Billerica, MA, USA).

### 2.10. Histological Quantification of Lipid Droplet Size in Epididymal Adipose Tissue

Epididymal adipose tissues were fixed in 10% neutral buffered formalin (GD CHEM, Seoul, Republic of Korea) for 3 days. Paraffin-embedded tissue sectioning and hematoxylin and eosin (H&E) staining were performed in the OBEN laboratory (Gyeongsan, Republic of Korea). The size of the lipid droplets in epididymal adipose tissue was calculated using ImageJ version 1.53a software (National Institute of Health, Bethesda, MD, USA).

### 2.11. Statistical Analysis

All data were analyzed using Prism software version 8 (GraphPad Software, Boston, MA, USA). Statistical significance was calculated using an unpaired Student’s *t*-test for comparisons between two groups, and using an ordinary one-way ANOVA, followed by Tukey’s multiple comparisons test, for comparisons involving more than two groups. A *p*-value < 0.05 was considered statistically significant, denoted as * *p* < 0.05, ** *p* < 0.01, *** *p* < 0.001.

## 3. Results

### 3.1. Discovery of L. plantarum GBCC_F0227, Which Effectively Catabolizes Triglycerides, through an In Vitro Screening Method

To identify novel probiotic strains capable of inhibiting fat absorption, we performed in vitro screening on lactic acid bacteria isolated from humans and various fermented foods. Each strain was cultured in MRS broth medium for 16 h, and then the remaining triglyceride content in the culture medium was quantified. As a reference, the MRS medium contained approximately 100 mg/dL of triglyceride. For comparative analysis, the following strains were used as controls: *L. plantarum* WCFS1 and *L. plantarum* KCTC 3108, *L. fermentum* KCTC 3112, *L. gasseri* KCTC 3163, and LGG. *L. plantarum* GBCC_F0227 and *L. plantarum* WCFS1 most effectively reduced triglyceride content (Figure 1A).

However, variations in growth among the different strains could have influenced triglyceride catabolism. To address this, the effective *L. plantarum* strains were cultured to OD_600_ values of 3 or 6, and then the triglyceride levels were measured to compare their catabolic activities. *L. plantarum* GBCC_F0227 exhibited the most pronounced triglyceride catabolism (Figure 1B). This contrasted with *L. plantarum* WCFS1’s effective catabolic ability, which can be observed in Figure 1A. These results demonstrate that *L. plantarum* GBCC_F0227, isolated from pickled cabbages, displays remarkable efficacy when catabolizing triglyceride.

### 3.2. Morphological, Genetic, and Physiological Characteristics of L. plantarum GBCC_F0227

*L. plantarum* GBCC_F0227 derived from fermented food exhibited cream-colored colonies in MRS agar culture and displayed a round and elongated rod shape similar to other lactic acid bacteria, as observed in scanning electron microscope images (Figure 2A).

When cultured in the MRS medium, *L. plantarum* GBCC_F0227 demonstrated robust growth levels under both aerobic and anaerobic conditions (Figure 2B), and also showed acid resistance comparable to that of LGG in anaerobic culture (Figure 3A,B). However, when incubated with artificial gastric fluid aerobically, the number of viable *L. plantarum* F0227 cells was lower than that of LGG, and this number was maintained when incubated with artificial intestinal fluid (Figure 3C). This shows that *L. plantarum* F0227 is more affected by acid than LGG in the presence of oxygen. Additionally, the safety of *L. plantarum* GBCC_F0227 was investigated through a cytotoxicity test using Caco-2 cells. Cytotoxicity was quantified using a lactate dehydrogenase (LDH) assay, which measures the LDH released from dying cells. The pathogenic strain *Staphylococcus aureus* ATCC 6538 showed cytotoxicity against Caco-2. In contrast, *L. plantarum* GBCC_F0227 showed minimal reactivity, demonstrating the safety of the strain (Figure 4).

Genomic analysis of *L. plantarum* GBCC_F0227 revealed the presence of nine replicons comprising one chromosome and eight plasmids, with a total genome size of 3,378,947 bp (Table 1). Of the total 3227 genes identified from *L. plantarum* GBCC_F0227, protein-coding genes, RNA genes, and pseudo genes constituted approximately 95.2%, 2.6%, and 2.1%, respectively. In particular, approximately 80.6% of genes exhibited predictable functions with no CRISPR region. The initial identification of *L. plantarum* GBCC_F0227 based on the 16S rRNA sequence was further validated through comparative genome analysis between strains for accuracy, showing 98.97% identity with the *L. plantarum* type strain DSM20174, reaffirming the classification of the strain as *L. plantarum* (Table 2).

### 3.3. α/β Hydrolase Genes with Lipase Activity Are Highly Expressed in L. plantarum GBCC_F0227

Through comparative genome analysis using the Lipase Engineering Database, we identified the enzyme genes in *L. plantarum* GBCC_F0227 involved in triglyceride catabolism. Specifically, these enzymes were abH04, abH08_1, abH08_2, abH11_1, and abH11_2, which belong to the α/β hydrolase family. As depicted in Figure 1B, *L. plantarum* GBCC_F0227 exhibited superior triglyceride catabolism compared to *L. plantarum* WCFS1. To ascertain whether *L. plantarum* GBCC_F0227 expressed these genes to a greater extent than *L. plantarum* WCFS1, qRT-PCR was performed at equivalent culture OD values. At an OD value of 3, the expression levels of *abH08_1*, *abH08_2*, and *abH11_2* were higher in *L. plantarum* GBCC_F0227 than in *L. plantarum* WCFS1. Similarly, at an OD value of 6, the gene expressions of *abH04*, *abH08_1*, *abH08_2*, and *abH11_1* were higher in *L. plantarum* GBCC_F0227 relative to *L. plantarum* WCFS1 (Figure 5). These results underscore *L. plantarum* GBCC_F0227’s superior triglyceride catabolism ability, attributed to the heighted levels of α/β hydrolases with lipase activity compared to *L. plantarum* WCFS1. Furthermore, our results highlight that the expression patterns of these genes vary depending on the growth stage of the strain.

### 3.4. Anti-Obesity Effects of L. plantarum GBCC_F0227 in HFD-Induced Mouse Obesity Model

Next, we investigated whether the efficient triglyceride catabolism of *L. plantarum* GBCC_F0227 translates to an anti-obesity effect when administered orally using an HFD-induced mouse obesity model. *L. plantarum* GBCC_F0227 not only significantly suppressed the body weight gain caused by the HFD, but also decreased the ratio of fat-to-lean body mass and the weight of epididymal adipose tissue (Figure 6A–C). Histological analysis further revealed a reduction in the size of the lipid droplets within adipocytes (Figure 6D,E). Moreover, *L. plantarum* GBCC_F0227 significantly upregulated the expression of adiponectin in adipose tissue (Figure 6F). Although *L. plantarum* GBCC_F0227 had no effect on the total cholesterol level, it significantly reduced blood triglyceride levels (Figure 6G). Together, these findings demonstrate the anti-obesity properties of *L. plantarum* GBCC_F0227, attributed to its capacity to inhibit fat absorption through triglyceride catabolism.

## 4. Discussion

Obesity and its associated metabolic diseases are becoming increasingly prevalent, prompting intensive research into preventive and therapeutic strategies. Recent attention has been directed towards the gut microbiome [23], owing to its significant involvement in the development of obesity [13,14,15]. In this context, we discovered *L. plantarum* GBCC_F0227, isolated from pickled cabbages, to be a promising candidate (Figure 1). This strain demonstrated remarkable efficacy when catabolizing triglycerides, a pivotal neutral fat abundant in HFDs. While *L. plantarum* WCFS1, previously reported to lower blood triglycerides in an HFD-induced mouse obesity model [42], also exhibited the ability to catabolize triglycerides, *L. plantarum* GBCC_F0227 surpassed its efficacy (Figure 1B). In addition, through whole-genome sequencing and comparative genome analysis using the Lipase Engineering Database, we identified specific enzyme genes, namely abH04, abH08_1, abH08_2, abH11_1, and abH11_2, belonging to the α/β hydrolase family with a lipase activity-related domain, as key players in triglyceride catabolism. The expression of these enzyme genes was elevated in *L. plantarum* GBCC_F0227 compared to *L. plantarum* WCFS1 (Figure 5), elucidating the basis of its superior triglyceride catabolism. Furthermore, *L. plantarum* GBCC_F0227 showed an anti-obesity effect in our 60 kcal% HFD-induced mouse obesity model, as evidenced by its ability to mitigate body weight gain, reduce blood glyceride levels, and diminish fat mass (Figure 6). *L. plantarum* GBCC_F0227 also increased the expression of adiponectin in adipose tissue, which is a key adipokine that increases insulin sensitivity and suppresses inflammation (Figure 6F). Considering its reduced expression in obesity, *L. plantarum* GBCC_F0227’s potential to increase adiponectin levels suggests a mechanism that mitigates inflammation and improves insulin sensitivity and blood glucose levels. In addition, *L. plantarum* WCFS1, which has lower triglyceride catabolism than *L. plantarum* GBCC_F0227, also suppressed increases in blood glyceride levels and body weight in the 40 kcal% HFD-induced mouse obesity model [31]. In the same experiment, *L. rhamnosus* LA68 also showed a similar suppressive effect on weight gain, but as it did not significantly lower blood glyceride levels, its mechanism of action is expected to be different.

For the industrial development of newly discovered gut microbial strains, mass production and safety assurance are imperative. Encouragingly, *L. plantarum* GBCC_F0227 showed robust and comparable growth characteristics under both anaerobic and aerobic conditions (Figure 2B), suggesting its potential for seamless mass production. In a cytotoxicity test using the LDH assay, *L. plantarum* GBCC_F0227 exhibited minimal reactivity, in contrast to the pathogen *Staphylococcus aureus* ATCC 6538, reaffirming its safety profile. Moreover, the acid resistance of *L. plantarum* GBCC_F0227 was similar to that of LGG, a widely used industrial strain. However, in the presence of oxygen, *L. plantarum* GBCC_F0227 had poorer acid resistance than LGG (Figure 3C), but given that *L. plantarum* GBCC_F0227 showed an anti-obesity effect in vivo, it seems that this property has little effect in these conditions. These favorable characteristics of *L. plantarum* GBCC_F0227 indicate its promising potential for industrial development. Recently, there has been a trend towards enhancing the business value of specific probiotic strains by demonstrating their efficacy not just in animal experiments but also in clinical trials [23,24,43,44]. For successful clinical trials, establishing detailed conditions for medium composition, culture conditions, and freeze-drying methods that can maintain optimal efficacy and stability is essential. Nevertheless, it is not easy to confirm the efficacy of probiotics in human trials. The efficacy of probiotics is generally confirmed in inbred mice under the same breeding conditions, but mouse experiments have limitations because the intestinal environments of these mice cannot reflect those of people affected by various environments, food habits, and genetic factors.

Our findings demonstrate that *L. plantarum* GBCC_F0227 exhibits an anti-obesity effect by efficiently catabolizing and reducing triglycerides, which are abundant in high-fat foods and serve as the primary cause of obesity. Taken together, these observations of *L. plantarum* GBCC_F0227 encourage the evaluation of its efficacy in humans through clinical studies and highlight its potential contributions to the prevention and treatment of obesity and associated metabolic diseases.

## Figures and Tables

**Figure 1 microorganisms-12-01086-f001:**
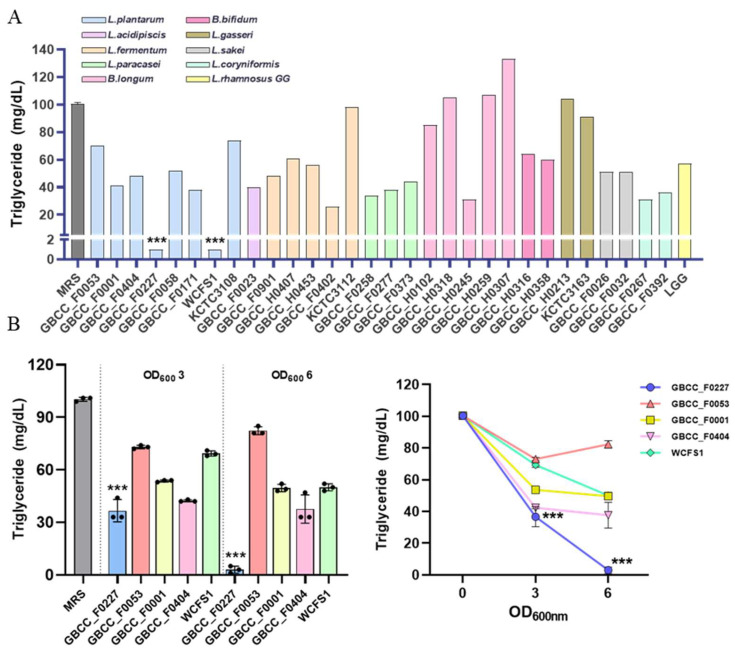
*L. plantarum* GBCC_F0227 demonstrates robust efficacy in triglyceride catabolism. (**A**) Lactic acid bacteria were anaerobically cultured in MRS medium at 37 °C for 16 h, and the triglyceride concentration in the medium was measured. (**B**) After culturing the lactic acid bacteria to OD_600_ values of 3 and 6, the triglyceride concentration in the medium was measured. Data are represented as mean ± SD (*n* = 3 per sample). *** *p* < 0.001 (one-way ANOVA, Tukey test).

**Figure 2 microorganisms-12-01086-f002:**
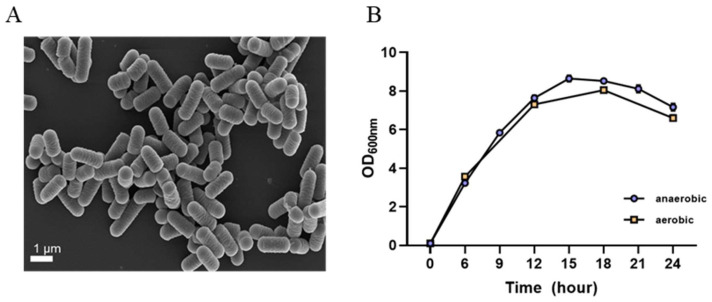
Morphological and cultural characteristics of *L. plantarum* GBCC_F0227. (**A**) Scanning electron microscope image. (**B**) Growth curve according to culture time under anaerobic and aerobic conditions. Data are presented as mean ± SD (*n* = 3 per sample).

**Figure 3 microorganisms-12-01086-f003:**
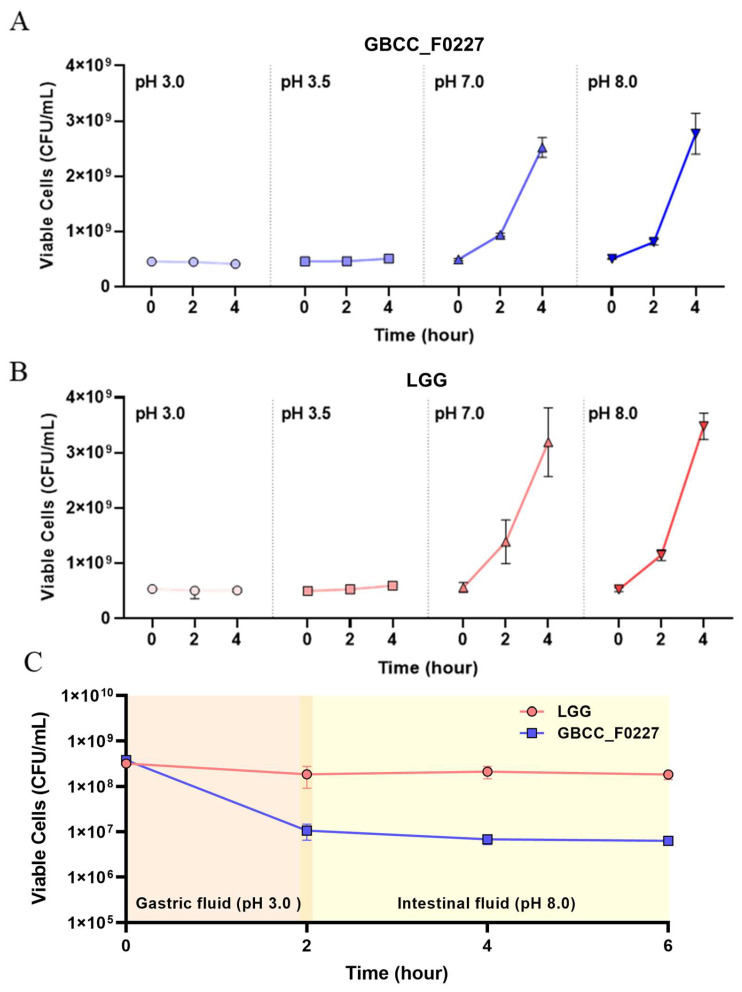
Comparison of *L. plantarum* GBCC_F0227 and LGG acid resistance. The numbers of viable cells (CFU/mL) of (**A**) *L. plantarum* GBCC_F0227 and (**B**) LGG after anaerobic culturing for 2 and 4 h under various pH conditions. (**C**) The numbers of viable cells of *L. plantarum* GBCC_F0227 and LGG over time in sequential incubation with artificial gastric fluid (pH 3.0) and artificial intestinal fluid (pH 8.0) under aerobic conditions. All data are presented as mean ± SD (*n* = 3 per sample).

**Figure 4 microorganisms-12-01086-f004:**
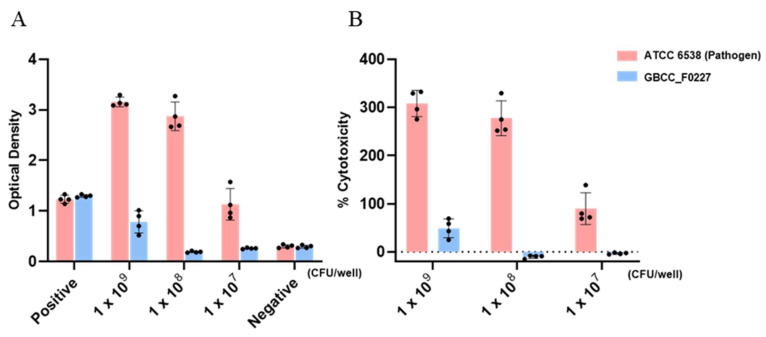
Analysis of *L. plantarum* GBCC_F0227 cytotoxicity. Caco-2 cells were incubated with *L. plantarum* GBCC_F0227 or *Staphylococcus aureus* ATCC 6538 for 24 h and cytotoxicity was measured using the LDH assay. The positive control was the lysis buffer, and the negative control was sterilized water. Cytotoxicity was expressed as (**A**) OD values and (**B**) % cytotoxicity. All data are presented as mean ± SD.

**Figure 5 microorganisms-12-01086-f005:**
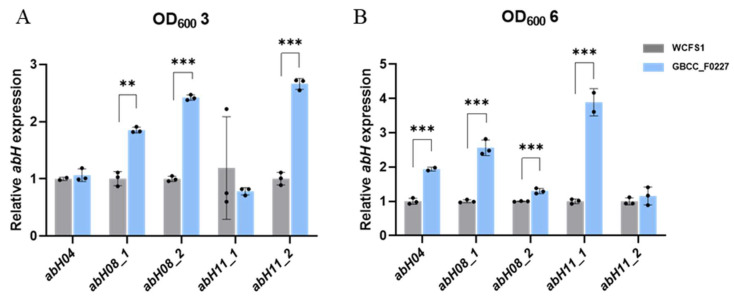
Expression patterns of genes encoding α/β hydrolases with lipase activity in *L. plantarum* GBCC_F0227. GBCC_F0227 and WCFS1 strains of *L. plantarum* were cultured to OD_600_ values of (**A**) 3 or (**B**) 6, respectively, and the relative expressions of genes encoding α/β hydrolases with lipase activity, such as abH04, abH08_1, abH08_2, abH11_1, and abH11_2, were measured using qRT-PCR. All data are presented as mean ± SD. ** *p* < 0.01, *** *p* < 0.001 (unpaired Student’s *t*-test).

**Figure 6 microorganisms-12-01086-f006:**
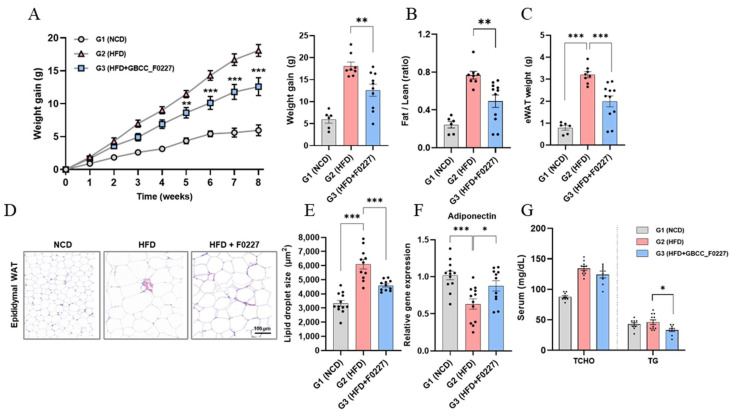
Effects of *L. plantarum* GBCC_F0227 in an HFD-induced mouse obesity model. Mice were fed either a normal chow diet (NCD), a 60% high-fat diet (HFD), or the HFD with oral administration of *L. plantarum* GBCC_F0227 at a dose of 5 × 10^9^ CFU/head daily. (**A**) Changes in body weight, (**B**) ratios of fat-to-lean body mass, and (**C**) weight of epididymal white adipose tissue (eWAT) were measured (G1 group, *n* = 6; G2 group, *n* = 8; G3 group, *n* = 11). (**D**) eWAT sections were stained with H&E and (**E**) the lipid droplet sizes of adipocytes were calculated. In the same adipose tissues, (**F**) the relative expression of adiponectin was measured using qRT-PCR (G1 group, *n* = 12; G2 group, *n* = 11; G3 group, *n* = 12). (**G**) The levels of total cholesterol (TCHO) and triglyceride (TG) in serum were measured (G1 group, *n* = 9; G2 group, *n* = 11; G3 group, *n* = 8). All data are presented as mean ± SEM. * *p* < 0.05, ** *p* < 0.01, *** *p* <0.001 (one-way ANOVA, Tukey’s multiple comparisons test).

**Table 1 microorganisms-12-01086-t001:** General features of *L. plantarum* GBCC_F0227 genome.

Features	Values	%
No. of replicons	9	-
Genome size (bp)	3,378,947	100.0%
DNA coding (bp)	2,808,715	83.1%
DNA G+C (bp)	1,498,741	44.4%
Total genes	3227	100.0%
Protein-coding genes	3073	95.2%
RNA genes	85	2.6%
rRNA genes	16	0.5%
tRNA genes	66	2.1%
Pseudo genes	69	2.1%
Genes with function prediction	2600	80.6%
Genes assigned to COGs	2325	72.1%
Genes with Pfam domains	2441	75.6%
Genes with signal peptides	168	5.2%
Genes with transmembrane helices	860	26.7%
CRISPR regions	0	-

**Table 2 microorganisms-12-01086-t002:** Average nucleotide identity among the species of *Lactiplantibacillus*.

Strains	Species	GBCC_F0227	DSM 20174	DSM 16365	DSM 10667	DSM 20314	TCF032-E4	LMG 26013	FI11369
GBCC_F0227	-	-	99.0	95.1	85.9	79.6	77.1	75.9	75.4
DSM 20174	*L. plantarum*	99.0	-	95.2	85.6	79.5	77.2	75.6	75.2
DSM 16365	*L. argentoratensis*	94.9	95.0	-	85.3	79.9	77.0	76.4	76.4
DSM 10667	*L. paraplantarum*	85.9	85.6	85.5	-	79.6	77.5	76.0	75.7
DSM 20314	*L. pentosus*	79.5	79.4	79.8	79.6	-	76.9	76.1	75.7
TCF032-E4	*L. herbarum*	77.1	77.1	76.9	77.5	76.7	-	75.9	75.5
LMG 26013	*L. xiangfangensis*	76.1	75.6	76.4	76.2	76.3	76.0	-	79.7
FI11369	*L. garii*	75.2	75.1	76.2	75.8	75.6	75.4	79.5	-

## Data Availability

Data are contained within the article.

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
