# Peer review of "Triglyceride-Catabolizing Lactiplantibacillus plantarum GBCC_F0227 Shows an Anti-Obesity Effect in a High-Fat-Diet-Induced C57BL/6 Mouse Obesity Model"

_microorganisms, 2024, doi:10.3390/microorganisms12061086_

Round 1

Reviewer 1 Report

Comments and Suggestions for Authors

Thank you for submitting the manuscript "Triglyceride-catabolizing Lactiplantibacillus plantarum GBCC_F0227 alleviates high-fat diet-induced obesity" to Microorganismos. The research isolated and characterized a potentially probiotic microorganism (and should have been designated as such since experiments on humans were not carried out). Overall, the manuscript needs extensive revision. Some notes:

 - I believe it is important to write the full name of the microorganism every time it appears in the text or create an abbreviation for the strain. What I don't find interesting is leaving it like this.

 - important to add in the title that the model is mice and what type of mice.

 - important to add in the title that the model is mice and what type of mice.

 - Line#19: "consumption of high-fat, low-dietary fiber" What? Diet, I suppose.

 - It is important to remember that the introduction must outline the objective of the work.

 - Lines#105-108: consider correcting the name of the microorganisms.

 - Line#103: consider including the number of animals used in each treatment.

 - consider including which fermented foods were used to isolate strains of microorganisms

 - The manuscript needs extensive revision of the English language and typographical errors.

 - the simple resistance to different pHs of the strain evaluated indicates possible resistance to the gastrointestinal tract but does not guarantee it. A simulation of the gastrointestinal system with enzymes and bile output would be necessary. This issue needs to be discussed in the text.

 - one of my concerns regarding the animal experiment is that a positive control (for example a commercial probiotic strain) was not included.

 - all in vivo work has limitations. Consider including them at the end of the discussion.

Comments on the Quality of English Language

The manuscript needs extensive revision of the English language and typographical errors. 

Author Response

Reviewer 1.

Thank you for submitting the manuscript "Triglyceride-catabolizing Lactiplantibacillus plantarum GBCC_F0227 alleviates high-fat diet-induced obesity" to Microorganismos. The research isolated and characterized a potentially probiotic microorganism (and should have been designated as such since experiments on humans were not carried out). Overall, the manuscript needs extensive revision. Some notes:

  1. I believe it is important to write the full name of the microorganism every time it appears in the text or create an abbreviation for the strain. What I don't find interesting is leaving it like this.

→ As the reviewer mentioned, all names of the microorganisms have been changed to their full names.

  1. important to add in the title that the model is mice and what type of mice.

→ As the reviewer commented, we have modified the Title.

  1. Line#19: "consumption of high-fat, low-dietary fiber" What? Diet, I suppose.

→ The typo mentioned by the reviewer has been corrected (Line 30 in the revised manuscript).

  1. It is important to remember that the introduction must outline the objective of the work.

→ As the reviewer commented, the purpose of the study has been added to Lines 83-86 in the revised manuscript.

  1. Lines#105-108: consider correcting the name of the microorganisms.

→ As the review said, incorrectly written names of microorganisms have been corrected (Lines 119-121 in the revised manuscript).

  1. Line#103: consider including the number of animals used in each treatment.

→ Mouse experiments were conducted several times, and the number of mice varied depending on the experiment, so the number of mice was already described in the legend of Figure 6 rather than in the Methods.

  1. consider including which fermented foods were used to isolate strains of microorganisms

→ As the reviewer commented, the name of the specific fermented food has been described in Lines 92, 271 & 373 in the revised manuscript.

  1. The manuscript needs extensive revision of the English language and typographical errors.

→ According to the reviewer’s comment, English correction was received from a company suggested by MDPI.

  1. the simple resistance to different pHs of the strain evaluated indicates possible resistance to the gastrointestinal tract but does not guarantee it. A simulation of the gastrointestinal system with enzymes and bile output would be necessary. This issue needs to be discussed in the text.

→ Based on the review’s comment, we added the results of an experiment using artificial gastric fluid and artificial intestinal fluid, and descriptions of this experiment are at Lines 141-150 in the revised Methods section, Lines 279-283 in the revised Results section, and Lines 404-407 in the revised Discussions section.

  1. one of my concerns regarding the animal experiment is that a positive control (for example a commercial probiotic strain) was not included.

→ As there were no direct comparison experimental results with the positive control strain, the reported experimental result of L. plantarum WCFS1, which has a lower triglyceride catabolism ability than L. plantarum GBCC_F0227, and Lactobacillus rhamnosus LA68 has been described at Lines 391-396 in the revised Discussion section. For reference, this experiment was conducted in a 40 kcal% HFD-induced mouse obesity model than the 60 kcal% HFD-induced mouse obesity model in while the efficacy of L. plantarum F0227 was confirmed (Lines 384-386 in the revised Discussion section).

  1. all in vivo work has limitations. Consider including them at the end of the discussion.

→ As the reviewer commented, A limitation of animal experiments has been added to Lines 413-417 in the revised Discussion section.

Reviewer 2 Report

Comments and Suggestions for Authors

Previous editor and authors, first of all I would like to thank you for the opportunity to review this work, which seems to me to be of high scientific quality, as it addresses all aspects of possible applicability to a new bacterial strain. The work seems to me that it can be published and I would only modify some presentation aspects:

- In Table 1, column 2, the % are left over after each figure.

- The p is lowercase in all cases, p<0.001 is correct, for example.

- The international hours abbreviation is h only.

- In Table 2, a single decimal must be placed in all values.

Author Response

Reviewer 2

Previous editor and authors, first of all I would like to thank you for the opportunity to review this work, which seems to me to be of high scientific quality, as it addresses all aspects of possible applicability to a new bacterial strain. The work seems to me that it can be published and I would only modify some presentation aspects:

  1. In Table 1, column 2, the % are left over after each figure.

→ It has been changed as the reviewer said.

  1. The p is lowercase in all cases, p<0.001 is correct, for example.

→ As the reviewer said, all ‘p’ has been changed to lowercase.

  1. The international hours abbreviation is h only.

→ As the reviewer said, all ‘hour’ has been changed to the international abbreviation 'h'.

  1. In Table 2, a single decimal must be placed in all values.

→ All values in Table have been changed to one decimal point.

Reviewer 3 Report

Comments and Suggestions for Authors

I appreciate the opportunity to review this manuscript. The authors conduct a very interesting study that addresses the need to identify probiotics to reduce obesity. In this particular case, produced by a high-fat diet.

The Introduction is confusing and lacks basic bibliographic references to understand the metabolic and immulogical mechanisms that are affected. The authors do not take into account other factors that can affect obesity, and which should be included. Lines 28 to 36, only one bibliographic reference is used, when statements are made that are understood to be based on rigorous empirical evidence.

In this sense, the references used are not up to date, even though there has been abundant literature in this area of study in recent years.

Lines 36 and 37. Is obesity only characterized by this?

The authors intend to introduce the dysbiosis present in obesity, without there being a connection that allows the reader to understand it.

lines 57 to 70. Again the authors make a direct association between obesity and dysbiosis, but other factors are not taken into account. This is a reductionist concept and a lack of knowledge in this area. In addition, studies in children with obesity indicate specific concentrations of bacterial families, which allows the identification of a metabolic profile. Some references of these studies are given below:

Bervoets, L., Van Hoorenbeeck, K., Kortleven, I., Van Noten, C., Hens, N., Vael, C., Goossens, H., Desager, K. N., & Vankerckhoven, V. (2013). Differences in gut microbiota composition between obese and lean children: a cross-sectional study. Gut Pathogens, 5(1), 10. https://doi.org/10.1186/1757-4749-5-10

Remely, M., Stefanska, B., Lovrecic, L., Magnet, U., & Haslberger, A. G. (2015). Nutriepigenomics. Current Opinion in Clinical Nutrition and Metabolic Care, 18(4), 328-333. https://doi.org/10.1097/MCO.0000000000000180

Other biomarkers such as C-reactive protein concentrations in obesity, lipid profile, butyric acid levels, bile acids, AMPK phosphorylation, etc. are also not taken into account....

The objectives and/or hypotheses of the study are not included.

The methodology seems adequate for the object of study. 

Results. Lines 227 to 229. What are the fermented foods used? Why were other strains of Lactiplantibacillus used? What were the selection criteria for comparison and how is it empirically justified that they may have similar or opposite effects?

The graphs are clear and facilitate the understanding of the results. 

In the discussion, there is hardly any reference to other studies, with this strain or with others. It is essential to be able to provide other results, both positive and negative, with the use of different bacterial families, phyla and strains. It is true that Lactiplantibacillus is one of the most studied, but there are many subspecies that have been studied, and other important strains, which are involved in metabolic regulation and improvement of the microbial profile.

Author Response

Reviewer 3

I appreciate the opportunity to review this manuscript. The authors conduct a very interesting study that addresses the need to identify probiotics to reduce obesity. In this particular case, produced by a high-fat diet.

  1. The Introduction is confusing and lacks basic bibliographic references to understand the metabolic and immulogical mechanisms that are affected. The authors do not take into account other factors that can affect obesity, and which should be included. Lines 28 to 36, only one bibliographic reference is used, when statements are made that are understood to be based on rigorous empirical evidence. In this sense, the references used are not up to date, even though there has been abundant literature in this area of study in recent years.

→ Based on the reviewer’s comments, new sentence has been added to Lines 34-36 of the revised Introduction section and references as empirical evidence have been added to Line 40.

  1. Lines 36 and 37. Is obesity only characterized by this?

→ This sentence has been replaced with another sentence in Line 40 of the revised Introduction section.

  1. The authors intend to introduce the dysbiosis present in obesity, without there being a connection that allows the reader to understand it. lines 57 to 70. Again the authors make a direct association between obesity and dysbiosis, but other factors are not taken into account. This is a reductionist concept and a lack of knowledge in this area. In addition, studies in children with obesity indicate specific concentrations of bacterial families, which allows the identification of a metabolic profile. Some references of these studies are given below:

Bervoets, L., Van Hoorenbeeck, K., Kortleven, I., Van Noten, C., Hens, N., Vael, C., Goossens, H., Desager, K. N., & Vankerckhoven, V. (2013). Differences in gut microbiota composition between obese and lean children: a cross-sectional study. Gut Pathogens, 5(1), 10. https://doi.org/10.1186/1757-4749-5-10

Remely, M., Stefanska, B., Lovrecic, L., Magnet, U., & Haslberger, A. G. (2015). Nutriepigenomics. Current Opinion in Clinical Nutrition and Metabolic Care, 18(4), 328-333. https://doi.org/10.1097/MCO.0000000000000180

Other biomarkers such as C-reactive protein concentrations in obesity, lipid profile, butyric acid levels, bile acids, AMPK phosphorylation, etc. are also not taken into account.

→ Based on the reviewer’s comments, the sentences in Lines 69-80 of the revised Introduction section have been changed and added.

  1. The objectives and/or hypotheses of the study are not included. The methodology seems adequate for the object of study.

→ As the reviewer commented, the purpose of the study has been added to Lines 83-86 of the revised manuscript.

  1. Results. Lines 227 to 229. What are the fermented foods used? Why were other strains of Lactiplantibacillus used? What were the selection criteria for comparison and how is it empirically justified that they may have similar or opposite effects? The graphs are clear and facilitate the understanding of the results.

→ The fermented foods used were various kimchi and pickled cabbages, and L. plantarum GBCC_F0227 was isolated from pickled cabbages. In Lines 92, 271 & 373 of the revised manuscript, it has been described that L. plantarum GBCC_F0227 was isolated from picked cabbages.

→ As triglyceride catabolism ability may vary depending on the bacterial species, we screened using various lactic acid bacteria species.

→ The selection criteria were productivity along with efficient triglyceride carbolic metabolism. In fact, it was confirmed that the selected strain not only showed the efficacy in HFD-induced mouse obesity model, but also had no problems with mass production.

  1. In the discussion, there is hardly any reference to other studies, with this strain or with others. It is essential to be able to provide other results, both positive and negative, with the use of different bacterial families, phyla and strains. It is true that Lactiplantibacillus is one of the most studied, but there are many subspecies that have been studied, and other important strains, which are involved in metabolic regulation and improvement of the microbial profile.

→ As the reviewer commented, animal experimental results of other strains with anti-obesity effects have been mentioned in Lines 391-396 of the Discussion section.

Round 2

Reviewer 1 Report

Comments and Suggestions for Authors

This reviewer thanks the authors who spared no effort to carry out all the revisions suggested by this reviewer and this greatly improved the quality of the manuscript

Reviewer 3 Report

Comments and Suggestions for Authors

The authors have implemented all requested modifications, resulting in a markedly enhanced manuscript.